# Flavonoids: Antiplatelet Effect as Inhibitors of COX-1

**DOI:** 10.3390/molecules27031146

**Published:** 2022-02-08

**Authors:** Cristina Zaragozá, Miguel Ángel Álvarez-Mon, Francisco Zaragozá, Lucinda Villaescusa

**Affiliations:** 1Pharmacology Unit, Biomedical Sciences Department, University of Alcalá, Alcalá de Henares, 28805 Madrid, Spain; francisco.zaragoza@uah.es (F.Z.); lucinda.villaescusa@uah.es (L.V.); 2Department of Medicine and Medical Specialties, Faculty of Medicine and Health Sciences, University of Alcalá, Alcalá de Henares, 28801 Madrid, Spain; maalvarezdemon@icloud.com; 3Ramón y Cajal Institute of Sanitary Research (IRYCIS), 28034 Madrid, Spain; 4Department of Psychiatry and Mental Health, University Hospital Infanta Leonor, 28031 Madrid, Spain

**Keywords:** flavonoids, antiplatelet activity, impedance aggregometry, cyclooxygenase (COX), arachidonic acid (AA), thromboxane B_2_ (TXB_2_), malondialdehyde (MDA)

## Abstract

Flavonoids are compounds with a benzopyranic structure that exhibits multiple pharmacological activities. They are known for their venotonic activity, but their mechanism of action remains unclear. It is thought that, as this mechanism is mediated by prostaglandins, these compounds may interfere with the arachidonic acid (AA) cascade. These assays are designed to measure the antiplatelet aggregation capacity of quercetin, rutin, diosmetin, diosmin, and hidrosmin, as well as to evaluate a potential structure−activity ratio. In this paper, several studies on platelet aggregation at different concentrations (from 0.33 mM to 1.5 mM) of different flavone compounds are conducted, measuring platelet aggregation by impedance aggregometry, and the cyclooxygenase (COX) activity by metabolites generated, including the activity of the pure recombinant enzyme in the presence of these polyphenols. The results obtained showed that quercetin and diosmetin aglycones have a greater antiplatelet effect and inhibit the COX enzyme activity to a greater extent than their heterosides; however, the fact that greater inhibition of the pure recombinant enzyme was achieved by heterosides suggests that these compounds may have difficulty in crossing biological membranes. In any case, in view of the results obtained, it can be concluded that flavonoids could be useful as coadjuvants in the treatment of cardiovascular pathologies.

## 1. Introduction

Many papers on the properties of flavonoids have been written, yet they remain under-recognized [1]. They have a peculiar benzopyran chemical structure (Figure 1), which is a prelude to activity, and research into their pharmacological properties is ongoing [2].

Flavonoids were introduced in therapeutics for their vitamin P properties, due to their ability to improve capillary permeability when there was any alteration in permeability [3]. This effect appeared to be controlled by prostaglandins. If prostaglandins controlled capillary permeability and flavonoids normalized it, it would be logical to suggest that these products would be interfering at a point in the arachidonic acid (AA) cascade, i.e., in the production of these prostaglandins (PG), so platelet aggregation is directly related to the production of AA metabolites [4,5].

The role of antiplatelet drugs is to prevent platelet activation either by inhibiting the platelet activation pathways or by stimulating inhibitory pathways [6]. These include cyclooxygenase (COX) blockers, as well as adenosine diphosphate (ADP) receptor antagonists and glycoprotein IIb/IIIa (GPIIb/IIIa) receptor antagonists [7].

The classification of flavonoids is quite broad and is carried out according to the constituents present in their chemical structure, and is divided into flavonols, flavones, flavanones, flavanols, isoflavones, anthocyanidins, dihydroflavonols, and chalcones [8]. These compounds play a very important role in a series of physiological and biochemical processes as they act as antioxidants [9], inhibitors of enzyme systems [10], and precursors of toxic substances and pigments [11], as well as being involved in the defense against infections [12,13], in the modulation of the inflammatory response, and of course in platelet aggregation [14].

Several compounds have been used in this paper, some known for their venotonic properties [15], with different flavone structures that could be the determinants of their pharmacological behavior [16].

The heteroside rutin and its aglycone quercetin are widely distributed in nature, in particular in the leaves of bearberry (*Arctostaphilos uva-ursi*, family *Ericaceae*), in the bulb of squill (*Urginea maritima*, family *Liliaceae*) in various species of the genus *Eucalyptus* and *Sophora*, as well as in buckwheat and elderberry, having been found to have a beneficial effect for lowering blood pressure and inflammation with 500 mg of the aglycone form [17]. Its antioxidant properties have also been described through lipoxygenase inhibition and metal chelating activity assays [18] and antiaggregant effects in vitro [19]. It has also been shown to modulate the production of pro-inflammatory cytokines [20] and to produce anti-cancer effects, including its ability to promote the loss of cell viability, apoptosis, and autophagy through the modulation of phosphatidylinositol-3-kinase/serine/threonine kinase/mammalian Target of Rapamycin (PI3K/Akt/mTOR), Wnt/-catenin, and mitogen-activated protein kinases/extracellular signal-regulated kinase 1/2 (MAPK/ERK1/2) pathways [21].

Other widely known polyphenolic compounds are diosmetin and its aglycone, diosmin. They have a flavonoid structure and are found in the leaves of the buchu tree (*Barosma* spp., family *Rutaceae*) and in the hyssop (*Hyssopus officinalis* L.), or synthetically derived from hesperidin [22], which is abundant in the citrus pericarp. These flavonoids are used as phlebotonics for venous insufficiency [16] and for alleviating the signs, symptoms, and severity of haemorrhoidal disease [23]. Diosmin has also showed a chondroprotective effect on human articular chondrocytes [24] and cardiovascular effects that support a beneficial role of this flavonoid on cardiovascular diseases (CVD), as well as the potential molecular targets involved [25]. This way, recent studies have evidenced that diosmin in combination with rivaroxaban reduces the risk of post-thrombotic syndrome after femoropopliteal deep vein thrombosis [26].

Despite having demonstrated their pharmacological activity in multiple physiological processes and their benefit for preventing the progression of certain pathologies, the mechanism of action of these flavonoids remains unclear. This paper aims at making a contribution to the antiplatelet properties of these drugs through impedance aggregometry as well as the way or pathway whereby they produce this effect, designing different experiments to find a potential COX blockage.

## 2. Results

### 2.1. Studies of Antiaggregant Activity by Impedance Platelet Aggregometry

AA and ADP were added to estimate 100% of platelet aggregation in both whole blood (WB) and platelet-rich plasma (PRP). Every flavonoid was assayed at 0.33, 0.83, 1.0, 1.25, and 1.5 mM concentrations. The results of aggregation (%) for flavonoids appear in Figure 2.

Platelet aggregation was inhibited by all of the flavonoids tested. This inhibition was more relevant in PRP samples with AA as a pro-aggregant agent (Figure 2B,D).

Heteroside flavonoids, rutin and diosmin, had an inhibitory effect amounting to 33.3% and 49.29%, respectively, in the presence of AA and PRP samples. The aglycones, quercetin and diosmetin, reached more effective results, namely, 71.51% for quercetin and 65.44% for diosmetin, at the same conditions; nevertheless, the heteroside hidrosmin achieved great data as aglycones (71.41%). The percentage of aggregation decreased as the concentration increased. There is a direct dose−response ratio. Thus, the highest results were obtained for quercetin, diosmetin, and hidrosmin at a 1.5 mM concentration in PRP samples with AA as the pro-aggregant agent (Figure 2D).

### 2.2. Studies of COX Activity by Measuring MDA Production

These studies were conducted using AA as the pro-aggregant agent and the PRP samples in view of the results obtained in the previous experiments (Figure 2D). The addition of AA without any flavonoid drug obtained a higher mark of malondialdehyde (MDA) production, which was considered to be a sample blank with a 100% of the COX activity. The flavonoid concentrations used were the same as in prior experiments. The positive control used in these assays was indomethacin at 0.00257, 0.006, 0.013, 0.019, and 0.02 mM concentrations (Figure 3).

Different concentrations of indomethacin showed a great decrease in COX-1 activity, reaching inhibition levels ranging from 52.73% at a lower dose to 87.71% at a higher dose, as authorized in therapeutics (Figure 3). Rutin and diosmin inhibited near 40% the COX-1 activity at the highest concentration of 1.5 mM. Diosmetin and hidrosmin inhibited more than 50% of the COX-1 activity, but the best marks were obtained using quercetin at 1.5 mM concentration, with inhibition being approximately 90% (Figure 3).

### 2.3. Study on Inhibition of Pure Human Recombinant Enzyme

These assays were designed to observe the activity of flavonoids on the pure human recombinant enzyme (h-COX-1). If the tested drugs were able to decrease the creation of some metabolites produced by COX, they would certainly inhibit the pure enzyme. The positive control used in these assays was indomethacin at 0.00257, 0.006, 0.013, 0.019, and 0.02 mM concentrations, and the concentration of flavonoids were the same as in prior experiments (Figure 4).

The positive control, indomethacin, exerted a greater decrease of h-COX-1 in this assay than the previous one, showing an inhibition of the pure enzyme at the higher concentration (0.02 mM), amounting to 96.51%. Heterosides showed better inhibition data than aglycones, with 73.42% of inhibition for rutin, 77.85% for diosmin, and 78.23% for hidrosmin, all of them at the highest concentrations (1.5 mM). Nevertheless, aglycones also decreased the activity of h-COX-1, but in a more moderate way than heterosides, with results amounting to 51.42% for quercetin and 28.14% for diosmetin.

### 2.4. TXB_2_ Levels as COX-1 Activity Indicator

This experiment was carried out in WB and calcium ionophore (CI) (25 mM) was added as the pro-aggregant agent. The addition of CI without any flavonoid drug obtained a higher mark of TXB_2_ production, which was considered as the sample blank with 100% of the COX-1 activity.

Again, indomethacin almost resulted in the total inhibition of the COX-1 activity. Aglycones of the flavonoid drugs, specially diosmetin at 1.5 mM, exerted an activity drop of the enzyme (70.29%), while the hidrosmin inhibition results amounted to 61.27%. The other heterosides also had this effect, but less dramatically—diosmin at 1.5 mM reached data of 31.23% of inhibition of enzyme activity, and 19.41% for rutin at 1.25 mM (Figure 5).

## 3. Discussion

The results of the impedance aggregometry assays, dealing with methods of platelet aggregation as described by Cardinal and Flower [27], show that all of the flavonoids under analysis can inhibit platelet aggregation to a greater or lesser degree. In general, the greatest degree of inhibition of aggregation is found when PRP and AA are used as a pro-aggregating agent, despite ADP, as recent authors have assayed [28]. Aglycones, quercetin and diosmetin, inhibit AA-induced aggregation in PRP to a greater extent than their corresponding glycosides, rutin and diosmin; however, hydrosmin, a diglycoside flavone (like diosmin), is found to exert a similar anti-aggregation activity to the aglycones. In all cases, the most relevant results are obtained at the highest concentrations tested (1.5 mM). The best data is obtained with diosmetin (20.1%), followed by quercetin and hidrosmin, both with very similar aggregation inhibition results of 23.49% and 23.80%, respectively.

A priori, it could be said that more apolar molecules, such as diosmetin and quercetin aglycones (without sugars in their structure), show a greater capacity to inhibit platelet aggregation, but the result obtained in the case of hidrosmin (diglycoside) is shocking, as it shows a capacity to inhibit aggregation that is closer to that shown by the aglycones studied.

Diosmin and hydrosmin differ only in the substituent at position 5. As for diosmin, this is a hydroxyl group, while in hydrosmin, the -OH group has been replaced by an HO-CH2-CH2-O- group. Given that both heterosides show very different results in terms of the inhibition of the platelet aggregation (in PRP assays, diosmin 42.23% and hydrosmin 76.25% with AA as an inducer of platelet aggregation, as well as diosmin 20.16% and hydrosmin 57.71% with ADP as an inducer of platelet aggregation), we can argue that the presence of such substitution in the case of hydrosmin is determinant for the activity.

In view of the results obtained in the impedance aggregometry assays, COX-1 activity studies by determining MDA production were performed using PRP samples and using AA as a pro-aggregating agent. In these studies, at the highest concentrations tested, the aglycones once again show a greater anti-aggregation capacity than the heterosides, with quercetin being the flavonoid that produces the greatest inhibition of the COX enzyme activity, reaching almost 90%. Some authors have written about the antiplatelet effect of quercetin [29,30], but there are no publications about any potential mechanism of action through the inhibition of COX. Diosmetin, the other aglycone under analysis, inhibits COX activity by 50%, a value very similar to that of hydrosmin, which, again, yields a different result to that of the other glycosides, inhibiting the enzyme activity by around 50% (like diosmetin). This result is in contrast with those achieved by rutin and diosmin, which inhibit the enzyme activity by 40% approximately.

At the lowest doses tested (0.33 mM), none of the heterosides inhibited the COX activity and, in the case of aglycones, inhibition was negligible. As for quercetin, the inhibition of the COX-1 activity is highly significant as the concentration increases.

Dimethyl sulfoxide (DMSO) is a great solvent for polyphenols, but its handling is complex working with platelets as their structure could be altered [31,32]; therefore, further concentrations of each flavonoid are difficult to obtain.

When pure enzyme inhibition studies of a recombinant origin are conducted, the results obtained show that all flavonoids are able to reduce the enzyme activity even at the lowest doses tested; however, in this case, the best values are obtained with heterosides. These assays are aimed at testing the degree of inhibition of the production of certain metabolites by flavonoids. The diosmin and hydrosmin heterosides show very similar results, inhibiting the h-COX activity by about 80%, followed by rutin (about 75%). With regard to aglycones, quercetin shows good enzyme inhibition results, although a very slow decrease in h-COX activity is found as the concentration increases. As for diosmetin, a dose increase does not result in an activity increase, with virtually identical results at doses of 1, 1.25, and 1.5 mM. These results are quite surprising compared to the obtained one in the others assays.

Quercetin, which had previously been shown to inhibit the enzyme activity by almost 90%, gave a result of about 60% when tested with the pure recombinant enzyme. However, all three heterosides, rutin, diosmin, and hydrosmin, show better results for inhibiting the pure enzyme. As for indomethacin (positive control), the results with the pure enzyme are much more relevant than those related to MDA production.

Studies to determine TXB_2_ levels are intended to indirectly measure COX activity following the addition of the flavonoids under the analysis of whole blood samples using IC as a pro-aggregating agent. In these assays, diosmetin aglycone exerted the strongest inhibition of COX-1 activity, being the most active at the 1.5 mM dose. Quercetin inhibits the COX activity by approximately 40%, while heteroside hydrosmin again shows a shocking result, inhibiting the enzyme activity by more than 60%, showing clear differences with other heterosides. Rutin and diosmin also inhibit the enzyme activity, but to a lesser extent than their corresponding aglycones, quercetin and diosmetin. As for rutin, 80.59% activity is achieved at a dose of 1.25 mM and with diosmin, the enzyme activity is 68.87% at a concentration of 1.5 mM.

The results of the studies show that aglycones generally show the greatest capacity to inhibit the enzyme activity and, therefore, the greatest antiplatelet capacity, although in the case of hydrosmin, some questions remain to be answered in order to understand its action.

Natural flavonoids, especially in their glycosylated forms, are the most abundant phenolic compounds found in plants, fruit, and vegetables [33]. Usually, the bioavailability of flavonoids is considerably restricted. After consumption, prior to absorption in the intestine or colon, microflora is able to hydrolyze glycosylated flavonoids [34]. Russo et al. explain how orally administered diosmin is quickly hydrolyzed into its aglycone, diosmetin, by enzymes from the intestinal microflora, and then be absorbed through the intestinal wall [35], this means, in vivo, diosmetin are quercetin are really the products that exert the effect.

As for heterosides, the best results are obtained when tested with the pure recombinant enzyme, which could suggest that these compounds have some difficulty in crossing biological membranes and gaining access to the interior of platelets. This could be justified by the presence of carbohydrates in its structure, which would increase their molecular size, as well as their water solubility. On the other hand, aglycones could more easily gain access to the interior of platelets and exert a greater effect on COX activity due to their smaller size. This would be in line with the results obtained for aglycones in the TXB_2_ production inhibition assays.

Flavonoids have shown to interfere with platelet aggregation by altering the metabolism of arachidonic acid, but each has its own spectrum of action on different pathways of platelet metabolism. These assays prove a blockage on the COX enzyme and, although the antiplatelet action of this large group of natural products has a difficult goal, further research is required. These results also conclude that flavonoids could be useful as antiplatelet coadjuvants in the treatment of cardiovascular pathologies. Their low bioavailability requires high doses in clinical use. Therefore, in our department, we are presently conducting research on a suitable vectorization system to optimize their behavior, just like with the micronized form of diosmin [36].

## 4. Materials and Methods

### 4.1. Materials

#### 4.1.1. Selected Drugs

The drugs used for these experiments are quercetin, rutin, diosmetin, diosmin, and hidrosmin (Figure 1). All of them are flavonoid drugs bought from Sigma-Aldrich (Sigma-Aldrich Chemical, Madrid, Spain) and the solvent, DMSO, from Dismadel (Dismadel, S.L., Madrid, Spain). The concentrations employed (0.33 mM, 0.83 mM, 1.0 mM, 1.25 mM, and 1.5 mM) were based on the therapeutical dose of Daflon^®^ [37].

#### 4.1.2. Volunteers

Eight healthy participants (six women and two men; aged 20.9 ± 1.3 (mean ± SD) years) were selected for the assays. The selected participants had not been on any antiplatelet treatment, or anovulants, or on any anti-inflammatory and antipyretic or steroid drug, in the last 6 months. Smokers were not allowed to participate, as well as people with any sign of disease.

#### 4.1.3. Ethical Approval

The study protocol was carried out in strict accordance with the guidelines of the 1975 Declaration of Helsinki, under approval of the Biomedical Ethics Committee of the University of Alcalá. Each volunteer had to sign written informed consent.

### 4.2. Blood Draws and Settings

Blood draws were carried out at the Haematology Service of the Principe de Asturias Hospital (Alcalá de Henares, Spain). Blood samples were obtained by antecubital puncture and were collected in 3.8% sodium citrate Vacutainer^®^ tubes (Dismadel S.L., Madrid, Spain). This anticoagulant was chosen because it has a smaller impact on the platelet pathways than other anticoagulants like heparin [38].

The following studies were carried out in whole blood (WB), in platelet-rich plasma (PRP), or both. Three pro-aggregant agents were used to induce platelet activation depending on the experiment: ADP (Sigma-Aldrich Chemical, Madrid, Spain) 5 µM [39], AA (Sigma-Aldrich Chemical, Madrid, Spain) 0.5 mM [40], and CI A23187 (Sigma Aldrich, Madrid, Spain) 25 mM [41]. The results vary according to the pro-aggregant agent used as they induce platelet aggregation through diverse ways.

#### 4.2.1. WB Samples

Blood extracted were homogenized and samples of 500 µL were allocated in aggregation Chronolog polyethylene buckets (Labmedics, Oxfordshire, UK), and 500 µL physiological saline solution (PSS) (Dismadel S.L., Madrid, Spain) were added to every bucket and they remained incubating at 37 °C for 1 h with every dissolution of flavonoid, or DMSO alone for the control. This incubation was performed in a thermostatic bath of Unitronic 320 Selecta (Tecnylab, Madrid, Spain) in order to improve solubility. This process took place no more than three hours after the blood draws.

#### 4.2.2. PRP Samples

PRP samples were obtained by gathering the supernatant from blood draws centrifugated in a Jouan B-3.11 centrifuge (Tecnylab, Madrid, Spain) for 10 min at 1200 rpm. Platelet counts were normalized to 200,000 platelets/µL PRP. PRP was solved in the hematologic solvent Diluid 601 (Biolab Diagnostics, Barcelona, Spain) for a short while and counting was carried out in a Neubauer cell chamber using a binocular microscope NIKON (Izasa S.L., Madrid, Spain). This procedure allowed for eliminating the hypothetical mistakes based on the automated cell counters, like suds categorized as particles. After this platelet count, the calculated volume was moved to the buckets to reach a final volume of 1ml adding PSS. They were incubated at 37 °C for 5 min with every dissolution of flavonoid, or DMSO alone for th econtrol.

### 4.3. Impedance Aggregometry

These assays were performed in a Chrono-Log 500 Lumi-Aggregometer (Labmedics, Oxfordshire, UK) linked to an Omnioscribe II data-logger (Instrumentos Testo, Cambrils, Barcelona, Spain) following the manufacturer’s instructions.

The samples only were handled with plastic materials. The experiment consisted of the measurement of the change in the electrical impedance between two electrodes when platelets were activated by pro-aggregant agents [42].

In this way, the electrodes submerged in the PRP or WB samples continuously stirred at 1200 rpm were covered by a platelet monolayer. Impedance remained constant without a pro-aggregant compound. Contrarily, adding a pro-aggregant compound induced the platelet adherence on the electrodes and resulted in greater impedance that was not able to be used as a measurement of the platelet aggregation.

### 4.4. Lipid Peroxidation Measurement

MDA results from the lipid peroxidation of polyunsaturated fatty acids. This is a process wherein several metabolites are produced when COX enzyme acts on AA [43]. Production of this metabolite has been used as a qualitative indicator of COX activity and platelet function measuring absorbance through spectrophotometry.

This assay was performed adding AA as the pro-aggregant agent, due to the release of MDA in presence of ADP being scarcer than in presence of AA, and the wavelength used was 532 nm to avoid other plasma pigments [44].

In order to extrapolate the release results of MDA, two calibration curves were performed at different concentrations of MDA, without pro-aggregant agent, in a UV/VIS Philips PU 8700 spectrophotometer (Inycom, Madrid, Spain). The samples were made in PSS as well as in PRP. The range of the calibration curves were 0.1–1 µM and 1–10 µM. Both curves showed regression coefficients near 1 (Figure 6).

The process to assay flavonoid samples was carried out in the same way. Once platelet aggregation of PRP samples with AA was registered and placed in propylene tubes, 375 µL of 40% trichloroacetic acid (ATA) (Sigma-Aldrich Chemical, Madrid, Spain) was added and covered to avoid oxidation. PSS was added up to a final volume of 2 mL, like in the samples used in the calibration curves. After centrifugation at 3500 rpm for 10 min, the supernatant was filtered through glass wool. One more centrifugation step was performed in similar conditions, and then 0.12 M tiobarbituric acid (TBA) (Sigma-Aldrich Chemical, Madrid, Spain) was added in a relation of 0.2 volume per volume of acid supernatant.

A water bath was used to incubate the covered tubes at 100 °C for 15 min. When the samples reached room temperature again, they were introduced into the spectrophotometer to measure absorbance at 532 nm.

The addition of AA without any flavonoid drug obtained the higher mark of MDA production, which was considered as the sample blank with 100% of the COX activity. Indomethacin (Sigma-Aldrich Chemical, Madrid, Spain)—the positive control—was used at different concentrations of 0.00257, 0.006, 0.013, 0.019, and 0.02 mM, taking into account that the clinical concentration is 0.004 mg/mL [45].

The flavonoids tested were dissolved in DMSO and different concentrations (0.33, 0.83, 1.0, 1.25, and 1.5 mM) to carry out these assays.

### 4.5. Study on Inhibition of Pure Human Recombinant Enzyme

This study was accomplished to determine the inhibitory effect of flavonoids on the pure human recombinant COX-1 enzyme. This one was purchased from Vitro S.A. (Madrid, Spain) and was provided in 10K phials performed in Tris-HCl 80mM, 1% Tween 20, and its purity reached 95%. Enzyme unit (EU) means the quantity of enzyme required to cause the transformation of 1 µmol of substrate per minute, measured at 610 nm. Phials stayed on ice and in a dark place while the experiment was conducted. The enzyme substrate used was AA, the pro-aggregant agent of the previous assays.

A chromogenic procedure stablished on the oxidation of *N*,*N*,*N*′,*N*′-tetramethyl-p- phenylenediamine (TMPD) while the PGG_2_ reduced to PGH_2_, determined the COX-1 activity [46]. AA was transformed into PGG_2_ in the platelet aggregation pathway, and this prostacyclin was immediately reduced to PGH_2_ by COX-1, thus, TMPD was proportionally oxidized to the enzyme activity.

Flavonoids solved in DMSO were tested at the same concentrations as in prior studies, 0.33, 0.83, 1.0, 1.25, and 1.5 mM, and were added to the wells, with 100 EU of pure enzyme and AA, the enzyme substrate. Indomethacin at 0.00257, 0.006, 0.013, 0.019, and 0.02 mM concentrations [45] was the positive control. Absorbance was obtained in a Biotek ELx800 Absorbance Microplate Reader (Izasa Scientific, Madrid, Spain) at a wavelength of 610 nm.

The experimental studies were carried out in solutions containing the pure enzyme incubated with the test compounds and AA as the enzyme substrate. Absorbance produced by TMPD was measured at 610 nm in a Biotek ELx800 Absorbance Microplate Reader (Izasa Scientific, Spain).

### 4.6. Enzyme Immunoassay for the Quantitative Determination of TXB_2_

TXA_2_ is another compound yielded from COX enzyme action on AA, which is quickly transformed to a steady metabolite called TXB_2_ [47]. Due to this steadiness, measuring the quantity of TXB_2_ produced was easier and more accurate than measuring the quantity of TXA_2_, thus, TXB_2_ was analyzed as amount of COX-1 activity [48].

Flavonoids solved in DMSO were tested at the same concentrations as in prior studies, 0.33, 0.83, 1.0, 1.25, and 1.5 mM, and in the same way, the positive control was indomethacin tested at 0.00257, 0.006, 0.013, 0.019, and 0.02 mM concentrations [45]. The pro-aggregant agent was CI A23187 (Sigma Aldrich, Madrid, Spain) at 25 mM concentration [48].

A specific enzyme immunoassay kit was used to estimate TXB_2_ production (TXB_2_ Biotrak Enzymeimmunoassay System, Amersham Biosciences, Little Chalfont, Buckinghamshire, UK), which holds an elevated acuity (0.2 pg) and the values of the standard curves ranged from 0.5 to 64 pg. Absorbance was measured using a Biotek ELx800 Absorbance Microplate Reader (Izasa Scientific, Madrid, Spain) connected to an automatic microplate washer Biotek ELx50 (Izasa Scientific, Madrid, Spain).

Flavonoid solutions assayed and DMSO as the control were added (2 µL) to the WB samples (1 mL) remaining at 37 °C for 1 h in a thermostatic bath (Unitronic 320 Selecta) (Izasa Scientific, Madrid, Spain). Afterwards, CI was adjoined to every tube and incubated for another 30 min. Upon completion of incubation, the tubes were introduced on dry ice to stop the reaction and introduced in a centrifuge Jouan 3.11 (Tecnylab, Madrid, Spain) for 10 min at 4000 rpm. The supernatant was reached to carry out the enzyme-linked immunosorbent technique.

### 4.7. Statistical Analysis

The results were formulated as the mean ± standard deviation (SD) of the data achieved for every assay. Because most variables did not fulfil the normality hypothesis, the Wilcoxon test was used to study the variance of the paired groups. The significance degree was set at *p* < 0.05. The statistical analysis was carried out using the SPSS-27.0 software (SPSS-IBM, Armonk, NY, USA).

## Figures and Tables

**Figure 1 molecules-27-01146-f001:**
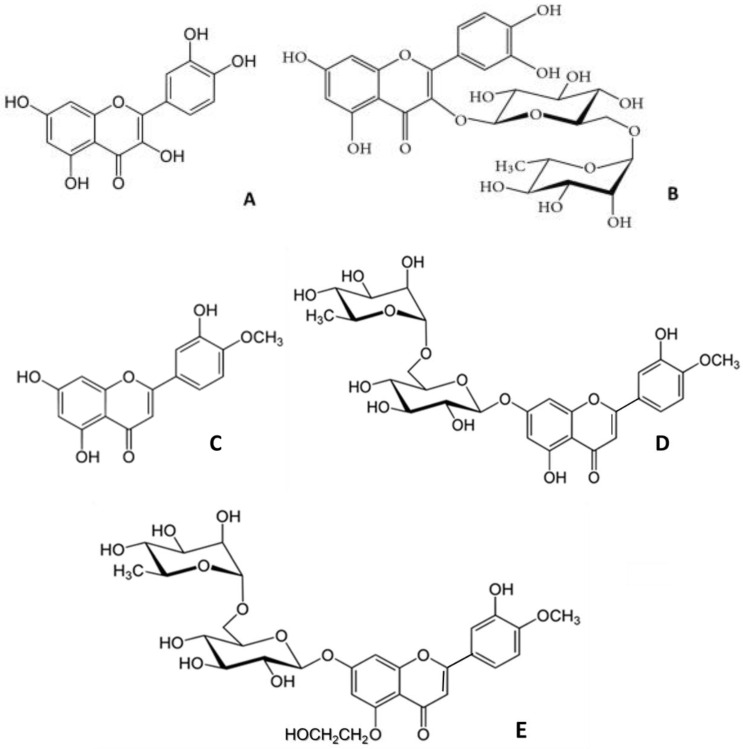
Flavonoids tested: (**A**) quercetin; (**B**) rutin; (**C**) diosmetin; (**D**) diosmin; (**E**) hidrosmin.

**Figure 2 molecules-27-01146-f002:**
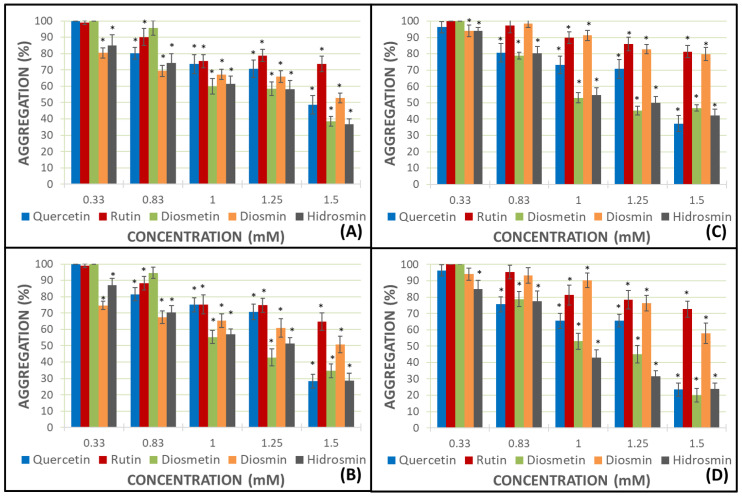
Platelet aggregation generated in the presence of each flavonoid in WB samples with ADP and AA ((**A**) and (**B**) panels, respectively), and in PRP samples with ADP and AA ((**C**) and (**D**) panels, respectively). Error bars indicate the standard deviation. Color bars describe the mean and standard deviation for *N* = 8. ** p* < 0.05: statistically significant variations in platelet aggregation between samples with and without the assayed flavonoid.

**Figure 3 molecules-27-01146-f003:**
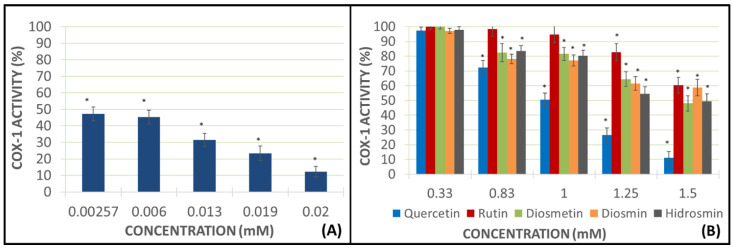
MDA production is represented as the COX-1 activity in the presence of indomethacin (panel (**A**)) and each flavonoid at different concentrations (panel (**B**)). Error bars indicate the standard deviation. Color bars describe the mean and standard deviation for *N* = 8. ** p* < 0.05: significant variations respecting COX-1 activity in presence or absence of the assayed compounds.

**Figure 4 molecules-27-01146-f004:**
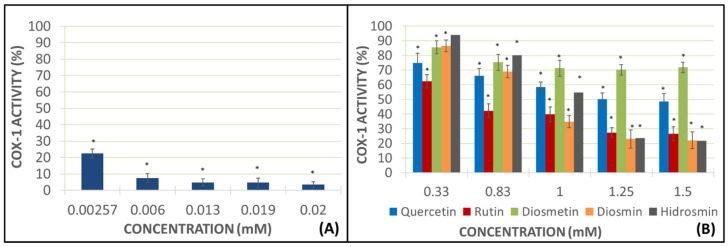
This diagram shows h-COX-1 works in presence of indomethacin as a positive control (panel (**A**)) and with different concentrations of each flavonoid (panel (**B**)). Error bars indicate the standard deviation. Color bars describe the mean and standard deviation for *N* = 8. ** p* < 0.05: significant variations of h-COX-1 in the presence or absence of the assayed compounds.

**Figure 5 molecules-27-01146-f005:**
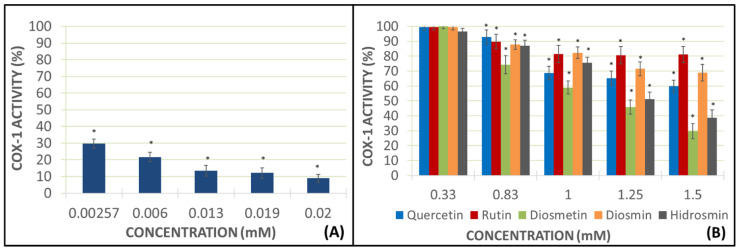
TXB_2_ production is represented as COX-1 activity in the presence of indomethacin (**A**) and each flavonoid at different concentrations (**B**). Error bars indicate the standard deviation. Color bars describe the mean and standard deviation for *N* = 8. ** p* < 0.05: significant variations respecting the COX-1 activity in the presence or absence of the assayed compounds.

**Figure 6 molecules-27-01146-f006:**
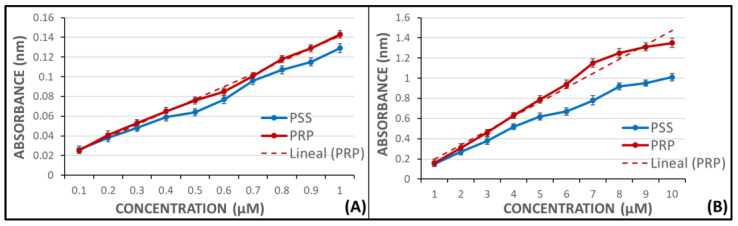
Calibration absorbance curves of MDA in PSS and PRP. (**A**) Range from 0.1 to 1 µM and linear fitting in PRP samples (*r* = 0.997). (**B**) Range from 1 to 10 µM and linear fitting in PRP samples (*r* = 0.994).

## Data Availability

Not applicable.

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
