# Peer review of "Flavonoids: Antiplatelet Effect as Inhibitors of COX-1"

_molecules, 2022, doi:10.3390/molecules27031146_

Round 1

Reviewer 1 Report

The manuscript is interesting and its results have a good relevance.
However, I believe that the discussion needs to be further worked on. The figures of the results should be better presented (they look like they were made in Microsoft Excel) and the results should be described more clearly and concisely. I would also like the authors to inform in the text perspectives on the potential use of these compounds and on their preclinical and clinical use in vivo

Reviewer 2 Report

Title

The needs to be formulated to include all manuscript aspects

Abstract

Indicate the abbreviation such as AA, COX for the first appearance, also the abbreviation in keywords, I suggest making abbreviations list.

 There are many linguistic mistakes i.e., in line 13, it should be “which exhibit pharmacological activities” instead of “which gives rise to multiple 13 pharmacological activities”., also, in line 16, use “this study aims to measure… instead of “These assays are aimed at”, generally the manuscript needs to linguistically check form an expert.

Introduction

The introduction sec is inclusive; however, the objectives of the study need to be cleared

Check language

in vitro must be italic in line 62

Indicate the abbreviations in lines 43, 65, 66

Materials and methods

Provide the model and origin of each material and device

Reformulate all the heads in this section

Clear the method of concentration preparation in lines 245-247

Include sec 4.1 and 4.2 under the main section of 4.1 materials as 4.1.1. drugs and 4.1.2. volunteers

Provide the ethical data in separate sec 4.1.3. ethical approval

Reformulate the label of the following secs

Indicate the abbreviations in line 263

Reformulate the head of 4.5 to lipid peroxidation measurement

Figure 6 can be converted to standard curve equation only

The authors once use mM and other nM for concentration, unify

Results

Reformulate Figures 3,4,5 or separate each Figure into two parts

Discussion

Add more recent citations in discussion

Where is the conclusion?

The references don’t follow the journal style, check

Check the outputs of all references

Reviewer 3 Report

In the manuscript the antiplatelet properties of flavonoids as inhibitors of COX-has been described. The investigations on biological  activity of compounds and on mechanism of their biological activity is an important scientific aspects. The relationship of the structure of flavonoids to their activity has also been discussed. In the manuscript data are clearly presented in corresponding Figures.

However, the advantages and novelty of the described investigations could be more emphasized especially compared with previously described.

Reviewer 4 Report

The manuscript ID molecules-1552479 entitled "Flavonoids: antiplatelet effect as inhibitors of COX-1" is a good study. Flavonoids are compounds with a benzopyranic structure, which gives rise to multiple pharmacological activities. They are known for their venotonic activity but their mechanism of action remains unclear. It is thought that, as this mechanism is mediated by prostaglandins, these compounds may interfere with the AA cascade. These assays are aimed at measuring the antiplatelet aggregation capacity of quercetin, rutin, diosmetin, diosmin, and hidrosmin, as well as at evaluating a potential structure-activity ratio. In this paper, several studies on platelet aggregation at different concentrations (from 0.33 mM to 1.5 mM) of different flavone compounds are conducted, measuring platelet aggregation by impedance aggregometry, COX activity by metabolites generated, including the activity of the pure recombinant enzyme in the presence of these polyphenols. The results obtained showed that quercetin and diosmetin aglycones have a greater antiplatelet effect and inhibit COX enzyme activity to a greater extent than their heterosides; however, the fact that greater inhibition of the pure recombinant enzyme was achieved by heterosides suggests that these compounds may have difficulty in crossing biological membranes. In any case, in view of the results obtained, it can be concluded that flavonoids could be useful as coadjuvants in the treatment of cardiovascular pathologies. 

I am curious, flavonoids are well-known antioxidants and anti-inflammatory agents with a huge volume of reports. But, none of them are used for regular practice to treat human diseases. Then, how the work will be promoting the treatment options to CVDs.?

Haggag et al. is an opinion article, they don't have any experimental evidence. Then, mentioning "hesperidin" as a COVID-19 drug is 
obscure info.

Round 2

Reviewer 1 Report

The manuscript has been satisfactorily improved. It can be considered for publication.

Author Response

Thank you very much

Reviewer 2 Report

I need to thank the authors for their kind replay to each point and the effort, which was made, therefore, the manuscript can be accepted for publication in Molecules

Author Response

Thank you very much.

Reviewer 4 Report

Thanks for the update. Yes, the use of flavonoids in clinical trials is limited by their bioavailability. The author enclosed some references and answers but it needs to be incorporated in the manuscript in the appropriate place. It is more important to promote small molecules for clinical use at present.  

Moreover, I am curious, how did the authors fix the low bioavailability of these flavonoids in the present study?. It seems they used raw-material from Sigma.
